# Satellite altimetry reveals spatial patterns of variations in the Baltic Sea wave climate

Nadezhda Kudryavtseva[1], Tarmo Soomere[1,2]

[1]Wave Engineering Laboratory, Department of Cybernetics, School of Science, Tallinn University of Technology, Tallinn, 12 618, Estonia
[2]Estonian Academy of Sciences, Tallinn, 10 130, Estonia

*Correspondence to*: Tarmo Soomere (soomere@cs.ioc.ee)

**Abstract.** The main properties of the climate of waves in the seasonally ice-covered Baltic Sea and its decadal changes since 1990 are estimated from satellite altimetry data. The data set of significant wave heights (SWH) from all existing nine satellites, cleaned and cross-validated against in situ measurements, shows overall a very consistent picture. A comparison with visual observations shows a good correspondence with correlation coefficients of 0.6–0.8. The annual mean SWH reveals a tentative increase of 0.005 m yr$^{-1}$, but higher quantiles behave in a cyclic manner with a timescale of 10–15 yr. Changes in the basin-wide average SWH have a strong meridional pattern: an increase in the central and western parts of the sea and decrease in the east. This pattern is likely caused by a rotation of wind directions rather than by an increase in the wind speed.

## 1 Introduction

The wave climate of relatively small semi-sheltered and seasonally ice-covered water bodies serves as a convenient indicator of changes in the atmospheric circulation and the related response of the water masses (Anderson et al., 2015). The associated changes in the wave fields are to a large extent controlled by the interplay of the geometry of the water body, changing winds and the overall reaction of the water masses. For instance, a reduction of the ice cover may strongly affect the wave loads (Tuomi et al., 2011; Ruest et al., 2016) and even small variations in wind properties may lead to marked changes in the wave fields. The analysis of wave properties makes it possible to reveal otherwise hidden changes in the forcing such as regime shifts (Soomere et al., 2015) and to highlight unusual reaction of the impacted environment, both direct (e.g., through an increase in severity of wave conditions, Wahl and Plant, 2015) and indirect (e.g., owing to a reduction in the ice cover (Orviku et al., 2003) or a change in the wave direction (Ashton et al., 2001)).

The limited size of such water bodies leads to large spatio-temporal variations in the wave properties. Thus, all sources of information about wave fields adequately reflect only wave conditions in the vicinity of the device or observation site. This feature is particularly significant in the Baltic Sea and similar water bodies that host extensive spatial variations in their wave climate (Soomere and Räämet, 2011, 2014). The related biases and uncertainties are generally recognized for several kinds of data sources such as visual observations from ships (Gulev and Grigorieva, 2006) or coastal locations (Hünicke et al.,

2015) but are often overlooked in the analysis of Acoustic Doppler Current Profiler (ADCP) or waverider data (Suursaar, 2015). Moreover, wave measurement devices are often removed well before the ice season (Tuomi et al., 2011).

Wave modelling provides a feasible option to complete the description of the wave climate and its changes. The major limitation is the wind quality. Even though reconstructed wind fields are adequate for the Baltic Sea, replication of the wave climate is still a major challenge for this basin (Tuomi et al., 2011, 2014; Hünicke et al., 2015). An intrinsic reason is the variability of ice cover extension from 12.5 % to 100 % of the sea surface (Leppäranta and Myrberg, 2009). Various decadal reconstructions show qualitatively similar patterns but reveal significant quantitative mismatches between different model outputs (Nikolkina et al., 2014) and between modelled and measured wave data.

Here we employ systematically the information about wave properties derived from satellite altimetry to quantify the wave climate of the Baltic Sea in terms of significant wave height (SWH). The relevant observations span today >20 yr and thus provide a valuable source for understanding the course and possible reasons for regional climate changes. Satellite altimetry provides homogeneous and continuous (along a certain line) data about the sea state over large areas in the open ocean (Hemer et al., 2010; Izaguirre et al., 2011; Young et al., 2011). Several recent studies have addressed the options for its use in the Arctic Ocean (Liu et al., 2016; Stopa et al., 2016). It, however, fails to provide information about low waves, is not applicable in the vicinity of land (Høyer and Nielsen, 2006) and is problematic for sea areas with high ice concentrations. It is thus not surprising that it has been only scarcely used to validate model results in the Baltic Sea (Cieślikiewicz et al., 2008; Tuomi et al., 2011). With certain precaution, it has shown good results for basins such as the Mediterranean Sea (Cavaleri and Sclavo, 2006; Galanis et al., 2012), a nearshore region of the Indian Ocean (Hareef Baba Shaeb et al., 2015; Patra and Bhaskaran, 2016), and for a range of regional seas such as the China Sea (Kong et al., 2016), the Arabian Sea (Hithin et al., 2015), the Chukchi Sea (Francis et al., 2011) or the German Bight (Passaro et al., 2015).

## 2 Data and methods

We employ the Radar Altimeter Database System (RADS) database (http://rads.tudelft.nl/rads/rads.shtml) (Scharroo et al., 2013). It provides data from nine satellites 1985–2015 (Table 1) that are uniformly reduced for multiple missions. Doing so diminishes the bias between data from different satellites and makes it possible to examine long-term changes in the wave climate. As several phenomena may substantially affect the quality of satellite altimetry, the consistency of the data has to be carefully checked. The relevant procedures, details about the missions and their temporal coverage as well as validation of the data against the records of waverider buoys and echosounder measurements are presented in Kudryavtseva and Soomere (2016). Here we provide only major aspects that are relevant for the subsequent analysis.

Different satellites have various densities of measurements. For example, SARAL and CRYOSAT-2 data fairly densely cover the entire Baltic Sea within each month (Fig. 1a) whereas, e.g., the POSEIDON data have very sparse coverage. We used flags provided by each mission (e.g., bad data because of rain or the presence of sea ice) as well as flags 7 and 11–13 in the RADS database that warned about possible large errors in range, backscatter coefficient, and SWH. The data with

backscatter coefficient >13.5 cdb (generally corresponding to wind speeds of <2.5 m s$^{-1}$) and large errors (>0.5 m, which is about half of the typical Baltic Sea wave height (Tuomi et al., 2011) in SWH normalized standard deviation (std) were excluded. The attitude flag 1 was applied to the GEOSAT data (Sandwell and McAdoo, 1988) and the zero values of SWH recorded by JASON-1 and ERS-1 were removed as likely erroneous ones. The GEOSAT Phase-1 data could not be validated

with the available buoy data and were excluded from the further analysis.

Possible biases of SWH for different missions were removed using cross-matching with in situ data and between satellites. The match of altimetry-derived and measured SWH is the best for offshore of the Baltic Proper due to a large number of data points. When compared with the in-situ data the cross-matched pairs showed bigger scatter for the wave-rider buoys and echosounders closer to the shore. For the in-situ measurement sites located less than 0.2° from the coast, the standard

deviation of cross-matches showed an increase by 14 %. However, the bigger scatter next to the shore does not affect the results of this study on condition that the data reflecting sea areas separated <0.2° from the coast were excluded from the analysis. The largest amount of matching altimetry and in situ data exists for JASON-1 (Fig. 1b). These sets reveal fairly small scatter, no systematic shift and a fairly small (around 4%) difference. TOPEX tends to overestimate the SWH by about 0.17 m whereas ENVISAT, ERS-1 and GEOSAT Phase 2 tend to underestimate the SWH by 0.15–0.23 cm. The bias is less

than 0.06 m for all other satellites. The bias between the SWH from different missions is consistent with the bias derived from comparison with the in situ data. For example, for CRYOSAT-2 and JASON-1 it is 0.04±0.06 m, and for ENVISAT and JASON-1 it is –0.19±0.03 m.

The data for the 1990s require additional handling. TOPEX had a switch in electronics in February 1999 which resulted in a shift in the time series. The TOPEX observations since February 1999 showed no bias with JASON-1. However, there is a

bias of 0.37 m between ERS-1 and TOPEX for 1991–1996. The buoy data revealed that the ERS-1 underestimated the SWH by 0.18 m and TOPEX overestimated it by 0.17 m in 1991–1996. Several satellites have significant drift in their SWH time series. The highest differences are between SARAL and JASON-2.

Even though part of the drift can be related to different frequencies of observations, some of the altimetry data sets eventually are not homogeneous in time. The absolute drift is reasonable (0.13 m for ENVISAT/JASON-1, 0.08 m for

SARAL/JASON-2 and 0.05 m for ERS-1/TOPEX) and does not affect the conclusions of this study. The overall corrections, based on the relevant linear regression analysis, are applied to the data: ERS-1 increased by 0.18 m, ENVISAT increased by 0.19 m and divided by 1.06, GEOSAT increased by 0.23 m, JASON-2 data divided by 1.009, TOPEX before February 1999 decreased by 0.17 m and after that divided by 1.014 (Kudryavtseva and Soomere, 2016).

Consistently with earlier analysis, a clear increase in scattering of buoy-satellite cross-matches was found for centroids of the

satellite snapshots separated <0.2° from the land. Such snapshots were excluded.

Large parts of the Baltic Sea are covered with ice each winter. Widespread variations in the extent of ice cover are an intrinsic and deeply nontrivial question in studies of wave climate of partially ice-covered sea areas. The presence of extensive ice cover may render the basic properties such as average wave height almost meaningless (Tuomi et al., 2011). For cross-matching the wave height data with the ice concentration measurements we used OSI-409-a dataset, taken from

EUMETSAT OSI SAF Global Sea Ice Concentration Reprocessing data (EUMETSAT Ocean and Sea Ice Satellite Application Facility. Global sea ice concentration reprocessing dataset 1978–2015, v1.2, 2015. Norwegian and Danish Meteorological Institutes, http://osisaf.met.no). We found that the presence of sea ice starts to affect the SWH values at as low concentrations as 10 %, and the wave heights are somewhat lower when ice concentration exceeds 30 %. The ice flag in the RADS data indicates the ice concentration >50 %. This threshold is appropriate for the Baltic Sea. Still, lowering it to the level of >30 % would result in the additional exclusion of only 0.1 % of the altimetry data (Kudryavtseva and Soomere, 2016).

## 3 Results

### 3.1 Observed changes and trends

The average SWH of the Baltic Sea in 1991−2015 is in the range 0.44–1.94 m (Fig. 2). As roughly one-third of calmer conditions are excluded from the analysis, the resulting values are clearly overestimated. For example, in a location in the Southern Baltic Proper where Soomere and Räämet (2014) (who underestimate SWH by about 15 %) report the average SWH ~0.7 m, the altimeter SWH is 1.294±0.002 m. The average SWH exceeds by 60–90 % the levels simulated in Soomere and Räämet (2014) and by about 30 % the levels reconstructed by Tuomi et al. (2011). The satellite-derived dataset, however, is expected to reflect adequately spatial patterns and temporal changes in the wave heights and the proportion of severe wave conditions.

Consistently with Tuomi et al. (2011), Soomere and Räämet (2014), Hünicke et al. (2015), the highest waves are found in the eastern and south-eastern Baltic Proper whereas the Gulf of Finland, Gulf of Riga and the (south-)western parts of the sea host much lower wave activity. The match of the frequency of satellite-derived and measured very rough seas is fairly good. Wave conditions with the SWH ≥ 4 m form 0.6 % of satellite snapshots in the entire sea whereas this proportion varies from 0.42 % to 1.4 % in various locations (Soomere, 2016). Extremely rough seas (SWH ≥ 7 m) occur twice (0.001 %) in the satellite data: 7.3 m on 31 January 1998 in the southern Baltic Proper and 7.4 m on 01 November 2001 in the northern Baltic Proper. Such wave conditions have been recorded less than 10 times in the entire Baltic Sea since 1978 (Soomere, 2016).

The major qualitative features of the basin-wide average SWH in 1991–2007 (Fig. 3) such as high wave activity in 1992 or 2007 and low wave activity in 1991, 1996 and 2006 match similar features of simulations based on geostrophic winds and ice-free sea (Soomere and Räämet, 2014) and results of visual wave observations (Soomere et al., 2015). However, there is a mismatch in the course of interannual variations and, more importantly, in the signs of trends in the average SWH established in the mentioned studies and those obtained in our analysis. Earlier analysis indicates an increase in the SWH and also in upper quantiles of the wave height in the open part of the Baltic Sea (Hünicke et al., 2015). These results are not confirmed by direct measurements (Broman et al., 2006; Soomere et al., 2012) and do not become visible in the outcome of fetch-based models (Suursaar, 2015). Also, the analysis of Soomere and Räämet (2014) show no significant trend in mean wave heights integrated over the entire Baltic Sea. Instead, the Baltic Sea wave fields exhibit extensive decadal-scale

variations that are out of phase in different sub-basins. In contrast, our study shows a tentative positive trend. It is worth to note, however, that the modelling results in partially ice-covered seas are affected by how the ice is treated in the modelling process (Tuomi et al., 2011; Ruest et al., 2016) and different runs often exhibit large discrepancy in the results (Nikolkina et al., 2014).

The number of single altimetry observations per year is <12 000 in 1991–1992, in the range of 20 000–31 500 in 1993–2001 and >30 000 since 2002 (Fig. 3d). This suggests that the data until 1992 are only conditionally usable to highlight long-term changes in the wave properties. For this reason, we only consider trends in the wave activity since 1993. The linear trend for 1993–2015 (based on 683 707 measurements) is 0.005 m yr$^{-1}$ at a 98 % significance level. It is consistent with a similar pattern in the 90th and 99th percentiles for the North Atlantic (Bertin et al., 2013) but is clearly weaker than analogous

trends extracted from numerical simulations using modelled wind fields (Hünicke et al., 2015) and disagrees with wave properties reconstructed using geostrophic winds (Soomere and Räämet, 2014). Consistently with Bertin et al. (2013), this trend is superimposed by marked interannual variability. As most of the increase is caused by variations in the 1990s, it does not necessarily mirror the long-term changes adequately. A similar trend in 2000–2015 is 0.002 m yr$^{-1}$. A large value of the relevant $p$-value ($p = 0.5$) signals that no significant changes in the basin-wide average SWH have occurred during the last

15 yr.

### 3.2 Testing the presence of trends

Following Wang and Swail (2001), an additional test to check the reliability of the whole-basin long-term trend in wave heights of 0.005 m yr$^{-1}$ was performed using the Mann-Kendall test since the least square fitting of a regression line is sensitive to gross errors and non-normality of the parent distribution (Sen, 1968). The Mann-Kendall test (Mann, 1945;

Kendall, 1955) is a nonparametric test for non-randomness of the data. The method is sensitive to the autocorrelation in the data (von Stork and Navarra, 1995). Therefore, a bootstrapping of the data was performed *("boot" R package, version 1.3.17)* with 5000 runs. For each run, a Mann-Kendall test was applied *("Kendall" R package, version 2.2)* and the results of the trial were averaged.

The average wave heights showed a significant autocorrelation of 0.4 with the lag of 1 and 2 years, whereas the higher

percentiles showed no evidence for the serial correlation. The Mann-Kendall test disproved the presence of a trend in the basin-wide wave height annual means, 90[th], and 99[th] percentiles (tau-statistics is 0.01, 0.03 and 0.005, correspondingly). This conjecture is consistent with the results of WAM simulations using geostrophic winds (Soomere and Räämet, 2014). We also checked the basin-wide annual means, 90[th], and 99[th] percentiles for each separate season and also did not find any significant trends with the bootstrapped Mann-Kendall test.

The scarcity of changes in the basin-wide SWH and 90[th] and 99[th] percentiles agrees with wave properties reconstructed using geostrophic winds (Soomere and Räämet, 2014). Interestingly, spatial variability of changes in these quantities is much weaker than similar variability in trends extracted from numerical simulations using modelled wind fields (Hünicke et al., 2015) and in the 90th and 99th percentiles for the North Atlantic (Bertin et al., 2013). The overall course of SWH is cyclic

with a timescale of 15–20 yr as suggested already by Broman et al. (2006). The higher percentiles of SWH reveal different patterns of temporal variation (Fig. 3a, b). Even though they exhibit formal increasing trends, the changes are not statistically significant and have either cyclic or sawteeth-like nature with markedly different time scales: 15–20 yr for the 75[th] percentile and much shorter, 3–4 yr for the 90[th] and 99[th] percentiles.

Simulations based on geostrophic winds indicated a complicated spatial pattern of statistically significant trends in the SWH in 1970–2007 (Soomere and Räämet, 2011). The largest increase in the SWH was identified near the Latvian coast of the Baltic Sea and a steep decrease in an area to the south of Gotland.

To assess where the changes appear locally in the Baltic Sea in our data set, we constructed maps of fitted linear trends for 1996–2015 (Fig. 4a). The presence and credibility of these changes can be to some extent evaluated based on the number of observations for each pixel used to calculate the trends (Fig. 4b). For the map of linear trends, the SWH values from single satellite altimetry snapshots were averaged for each time interval over a grid of 40×40 pixels, each pixel 0.4×0.3°. Only pixels that involve ≥8 recordings are taken into account to avoid errors due to false averaging. The map of number of observations per each 40×40 pixels (Fig. 4b) shows that the number of satellite measurements is almost uniformly distributed across the Baltic Sea with systematically fewer observations next to the coast. Therefore, if trends similar to these in the western part were also present in the eastern part of the Baltic Sea, they should have been visible in the satellite altimetry data set.

Even though a certain increase in the wave activity has occurred in almost entire sea (Fig. 4a), the changes in the SWH reveal a strong spatial pattern. The changes are minor or indistinguishable in smaller sub-basins. An increase in the SWH in areas open to the North Sea is apparently connected with an increased level of storms and swells in the North Atlantic (Bertin et al., 2013). Importantly, a significant increase is observed in the western sections of the northern Baltic Proper, to the southwest of Gotland and in the south-western part of the sea. An application of the Mann-Kendall test showed similar results (Fig. 5a). A decrease is observed in many locations near the eastern coast of the sea. Even though these changes are not directly comparable with the outcome of numerical simulations for single decades (Soomere and Räämet, 2014), they show a pattern that radically differs from the usual perception of a gradual increase in the wave activity in the eastern Baltic Sea (Hünicke et al., 2015) and has also clear mismatch with trends established for 1970–2005 (Soomere and Räämet, 2011).

To rule out that the observed east–west pattern of trends can in principle be caused by inhomogeneity of observations in time and space, a simulated dataset was created as

$$H_S = 0.005 \text{ m yr}^{-1} \, t + b + err, \tag{1}$$

where $H_S$ is the simulated significant wave height, $b = -10.39$ is the intercept, and $err$ is a normally distributed random error with a standard deviation of 0.5 m. The $H_S$ data were selected for exactly the same locations and times as the real dataset. Then the same data analysis procedures were applied to the simulated dataset. The results are shown in Fig. 5b on the same 40×40 pixels grid as in Fig. 4. Significant trends are retrieved both with linear regression and Mann-Kendall method for the

whole Baltic Sea basin, suggesting that time and space inhomogeneity in data could not result in an observed east-west pattern of trends.

To additionally check the possible masking effect of short-term fluctuations in the wave heights (Fig. 3), we applied the described analysis for three shorter consecutive periods: 2009–2011 (Period 1), 2011–2013 (Period 2), and 2013–2015 (Period 3). There was a substantial increase (up to 0.73 m, equivalent to $4\sigma$ (fourfold std) variation, the false positive rate is 0.003 pixels per map) in the SWH between Period 1 and Period 2 in the northern Baltic Proper (Fig. 6a). During the subsequent Periods 2 and 3, the SWH almost reverts to the level in Period 1. The maximum decrease is 0.70 m (std is 0.17 m, which shows a difference of $4.1\sigma$, the false positive rate at a $4\sigma$ level is 0.01 pixels per map).

The established patterns of variations in the SWH in Fig. 4 suggest, in contrary to numerical simulations described in (Hünicke et al., 2015), that a very slow increase or even a decrease in the SWH has occurred along the eastern coast of the sea. As no long-term instrumental wave record exists for this region (Hünicke et al., 2015), we check this result against visually observed wave properties near the Latvian coast (Fig. 1a). Data sets with acceptable quality are available at two sites: Ventspils (57.4°N, 21.53°E) and Liepaja (56.47°N, 21.02°E) until 2011 (Soomere, 2013). The yearly averages of visually observed SWH show a fairly good qualitative correspondence with satellite altimetry data within 0.3° from these sites (Fig. 7). The correlation coefficient between annual average SWH from the two data sets is $r = 0.61$ at Liepaja, based on 10 yr of coinciding data (2002–2011) and 444 cross-matching pairs in total. At the Ventspils site (where the data quality of visual observations is lower, Soomere, 2013) the correlation is $r = 0.81$ in 2009–2011, with 354 pairs. The lower correlation coefficient between the satellite altimetry data and visual observations can be a result of a lower number of matched pairs at this location (44.4 pairs/yr in Liepaja and 118 pairs/yr in Ventspils on average). To test the effect of the grid cell size on the correlation coefficients, we have checked the cross-matches within 0.2° and 0.1° from Liepaja and Ventspils observation sites but did not find a significant change in the correlation coefficients. For example, a reduction of the grid cell to 0.2° almost does not affect the correlation coefficient for Liepaja (0.60, 9 years of coinciding data) and improves the formal correlation for Ventspils (0.93, 3 years of coinciding data).

## 4 Discussion and conclusions

We estimated for the first time the main properties of wave climate of the entire Baltic Sea based on two decades of satellite altimetry records. Even though the resulting values of the long-term average SWH are biased towards higher waves (because records of low wave conditions are systematically ignored), the estimated average SWH reasonably matches the outcome of numerical simulations and in situ records. The long-term SWH over the entire sea exhibits a statistically significant increase by 0.005 m yr$^{-1}$. However, the presence of such a trend was not confirmed by the non-parametric Mann-Kendall test. A possible reason for the failure of this test is that the trend in question is superimposed on marked interannual variations (consistently with the overall pattern of exceptionally high interannual variability in the winter over the entire North Atlantic

(Woolf et al., 2002)) and a cyclic course of the overall wave activity and higher quantiles of SWH on a time scale of 15–20 yr.

The commonly accepted reasons behind the possible increase in the Baltic Sea wave heights are i) a reduction of sea ice in northern parts of the sea and ii) an increase in the wind speed (Hünicke et al., 2015). Both these reasons should lead to a spatially inhomogeneous increase in the wave heights, first of all in seasonally ice-covered northern part of the Sea and along the eastern segments of the basin where the predominant south-westerly and north-north-westerly winds usually create the severest wave conditions. Our analysis reveals an unexpected strong meridional pattern of changes: the wave heights have increased in the western offshore of the sea and have decreased (or exhibit no changes) along the eastern nearshore. It is, therefore, unlikely that a discernable increase in the wind speed has occurred in this region. This among other things means that a greater level of storms and swells (Bertin et al., 2013) may only characterize some parts of the North Atlantic. This is consistent with the conclusion that the basin-wide average geostrophic wind speed has not increased over the entire Baltic Sea (Soomere and Räämet, 2014).

The established meridional pattern of trends in the wave heights does not reflect the variations in the density of altimetry observations as shown by the synthetic trend test (Fig. 5b). Also, the short-term changes on the timescale of ~3 yrs are located in the Baltic Proper (Fig. 6a, Fig. 6b), where the density of the observations is the highest. It is also not likely that the identified patterns represent systematic errors in the data set. The data used in the study were carefully cross-validated and all low-quality data excluded from the analysis. This concerns as measurements in the nearshore locations (<0.2° from the land), sea areas with ice concentration >30 %, backscatter coefficient >13.5 cdb, and all entries with errors >0.5 m in normalized SWH (see Kudryavtseva and Soomere, 2016 for more details). An implicit support to the validity of the established trends and their spatial patterns is a high correlation between visually observed wave properties and SWH derived from satellite altimetry for locations directly offshore from the coastal observation sites.

In this context, it is remarkable that no statistically significant trend in the SWH can be identified for the south-eastern part of the Baltic Sea. This area evidently provides the most reliable altimetry data in the Baltic Proper. This vast region of the open sea has a relatively straight coastline. It hosts no islands and ice cover appears infrequently. It is natural to expect that any systematic change in the wind speed would generate an associated change in the wave heights in some part of this region. Therefore, the likely reason for the discussed pattern of changes is a rotation of the direction of moderate and strong winds. This alteration is driven by major changes in the spatial distribution of low-pressure systems over the North Atlantic and Arctic Ocean (Lehmann et al., 2011).

Even though wind direction is often considered as a secondary parameter in climate studies, the impact of rotation of wind patterns on wave fields is a generic issue (Hemer et al., 2010). This effect is comparatively strong in regional seas and large lakes with an elongated shape (Niu and Xia, 2016) where veering of the prevailing winds may lead to large changes in the fetch length. The presented results suggest that changes in wind direction may serve as the predominant driver of regional climate changes in such water bodies (Anderson et al., 2015).

**Data availability**

The satellite altimetry data are retrieved from the Radar Altimeter Database System (RADS) database (http://rads.tudelft.nl/rads/rads.shtml) (Scharroo et al., 2013). Annual observed wave heights in Fig. 7 are extracted from Soomere (2013).

**Author contribution**

N. Kudryavtseva performed the analysis of satellite altimetry data, wrote sections that address this analysis and prepared most of the images. T. Soomere compiled literature overview and developed the comparison of altimetry data with existing numerical simulations and visual observations. The Results and Discussion section were written together.

**Acknowledgements**

The research was initiated by the Small Grants Scheme Project „Effects of climate changes on biodiversity in the coastal shelves of the Baltic Sea" 2015–2016 (European Economic Area (EEA) grant No. 2/EEZLV02/14/GS/022) and supported by the institutional financing by the Estonian Ministry of Education and Research (Grant IUT33-3) and the ERA-NET+ RUS project EXOSYSTEM (Grant No. ETAG16014).

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

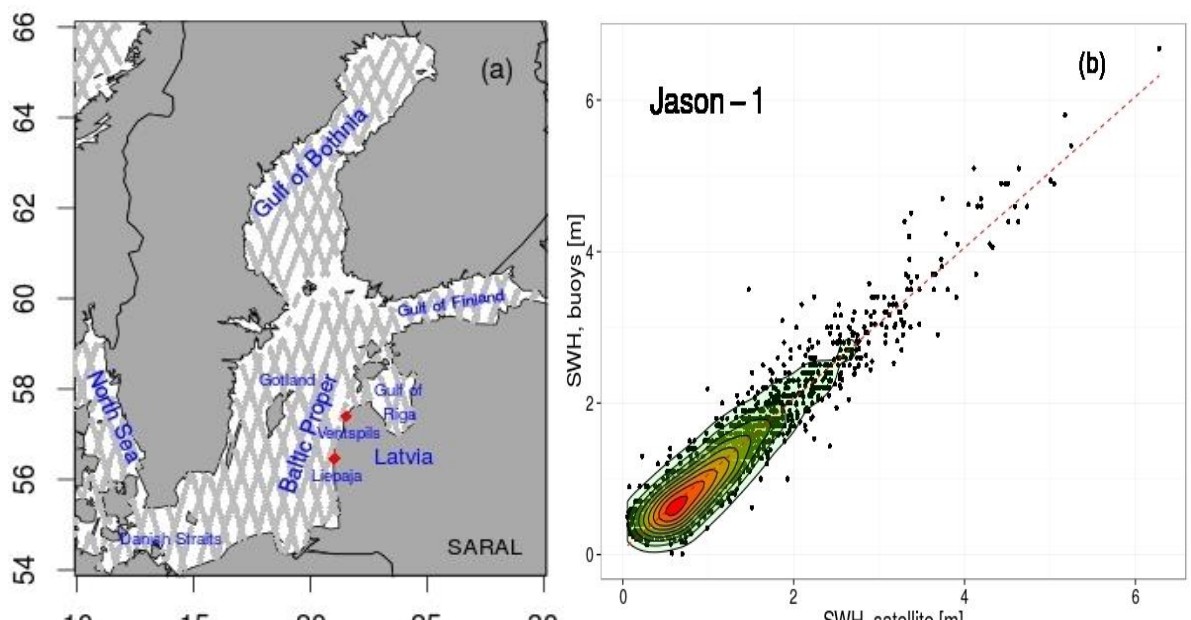

**Figure 1: Validation of satellite altimetry data, (a) an example of monthly visits of SARAL satellite to the Baltic Sea region, (b) a comparison of SWH derived using JASON-1 and in situ data. Adapted from Kudryavtseva and Soomere (2016).**

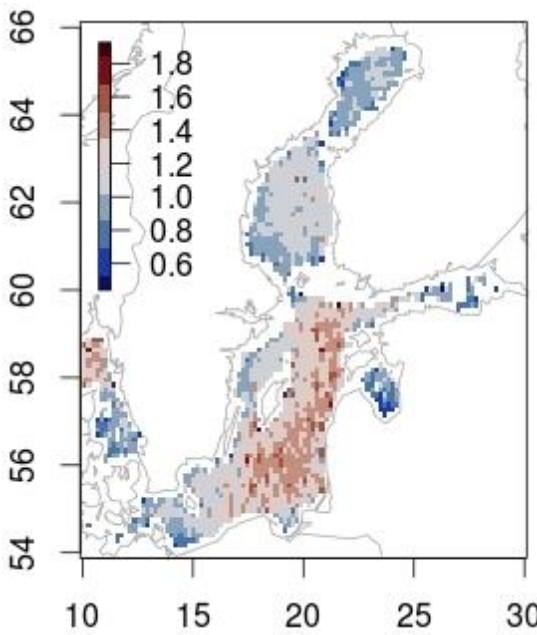

**Figure 2:** Average significant wave height [m, color scale] in the Baltic Sea in 1993–2015 from satellite altimetry. The approximate pixel size is 0.2×0.1° (100×100 pixels grid).

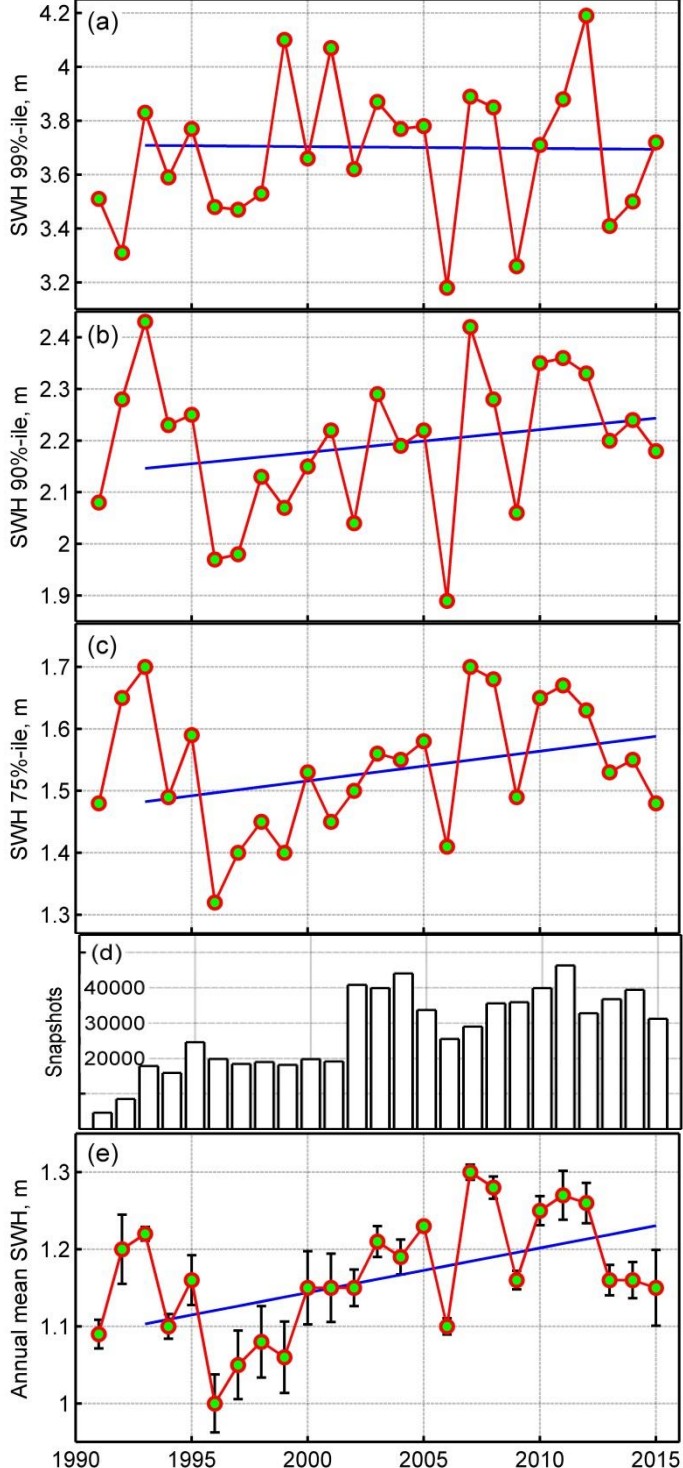

**Figure 3: Wave climate derived from the satellite altimetry in terms of the annual mean SWH in the entire Baltic Sea (e), 75th, percentile (c), 90th percentile (b), and 99th percentile (a) of single SWH records. The straight lines show the linear trends (regression line fitted to 1993–2015). The small panel (d) indicates the count of employed altimetry snapshots.**

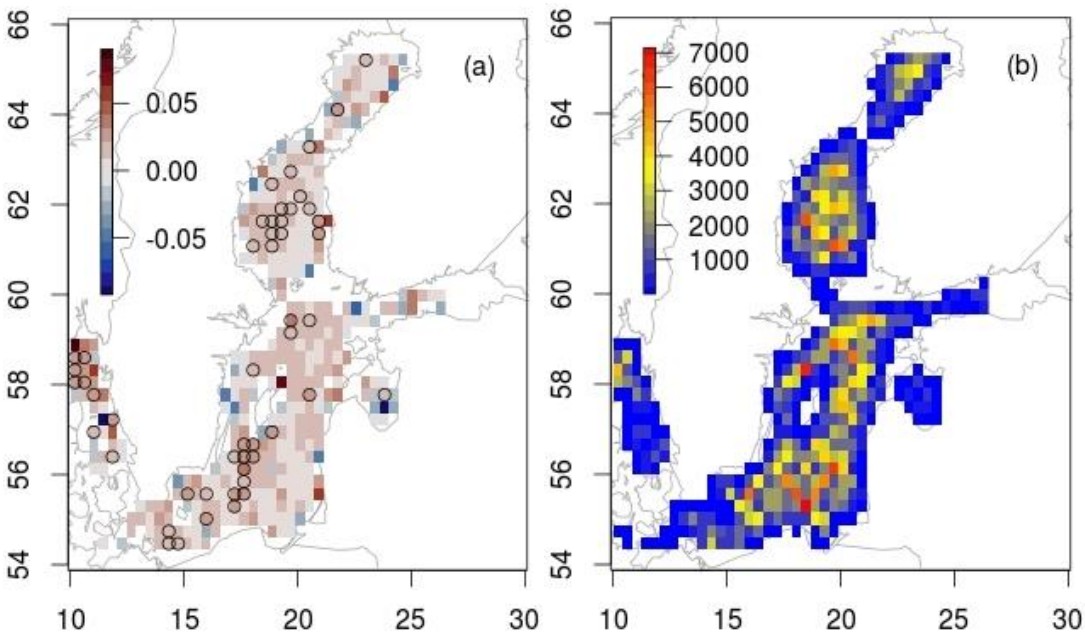

5  **Figure 4: Spatial variability of wave climate in the Baltic Sea, (a) Slope (m yr⁻¹, color scale) of linear trends in significant wave heights in the Baltic Sea in 1996–2015 from altimetry measurements (pixel size 0.4×0.3°). Crosses indicate the pixels hosting a statistically significant (at a >99 % level) increase or decrease in the SWH, (b) Number of observations per pixel in 1996-2015 (pixel size 0.4×0.3°). An increase in the pixel size compared to Fig. 2 (where the average SWH height is plotted using all available data over the whole period of observations) makes it possible to get a reasonable number of data points for each year for each**
10  **particular pixel.**

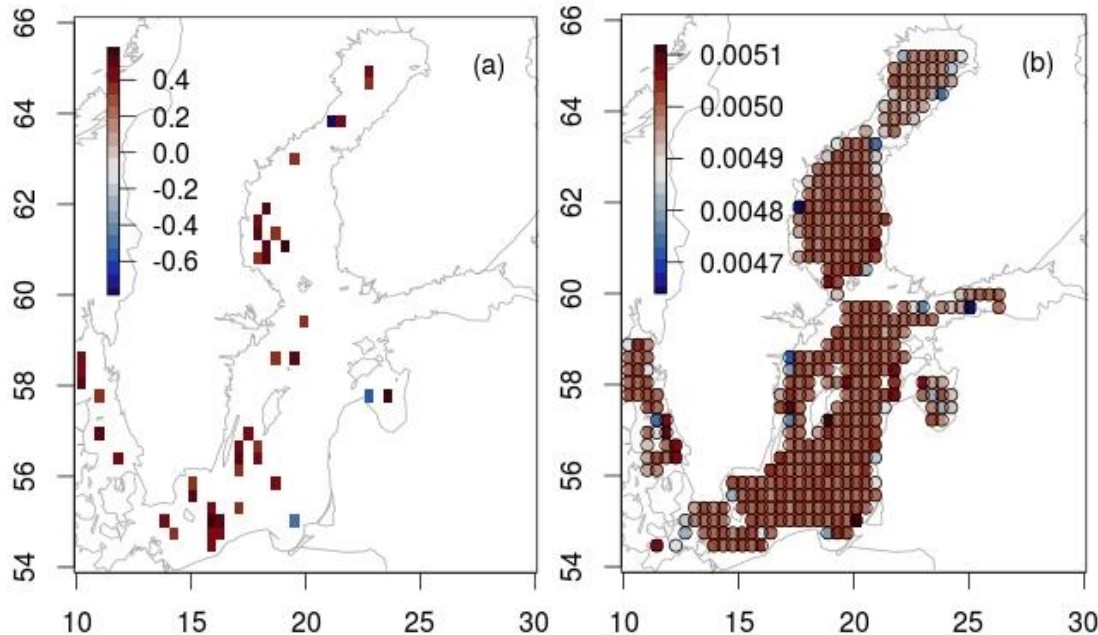

**Figure 5: Spatial variability of wave climate in the Baltic Sea, (a) Mann-Kendall tau statistics of trends in SWH in the Baltic Sea in 1996–2015, winter months, from altimetry measurements (pixel size 0.4×0.3°). The data are shown only for statistically significant (at a >95 % level) increase or decrease in the SWH, (b) Trends found in synthetic data generated for the same positions and times as the studied satellite altimetry dataset in 1996-2015 (pixel size 0.4×0.3°). The data were created to uniformly follow a positive trend of 0.005 m yr$^{-1}$. Crosses indicate statistically significant trends (at a >99 % level).**

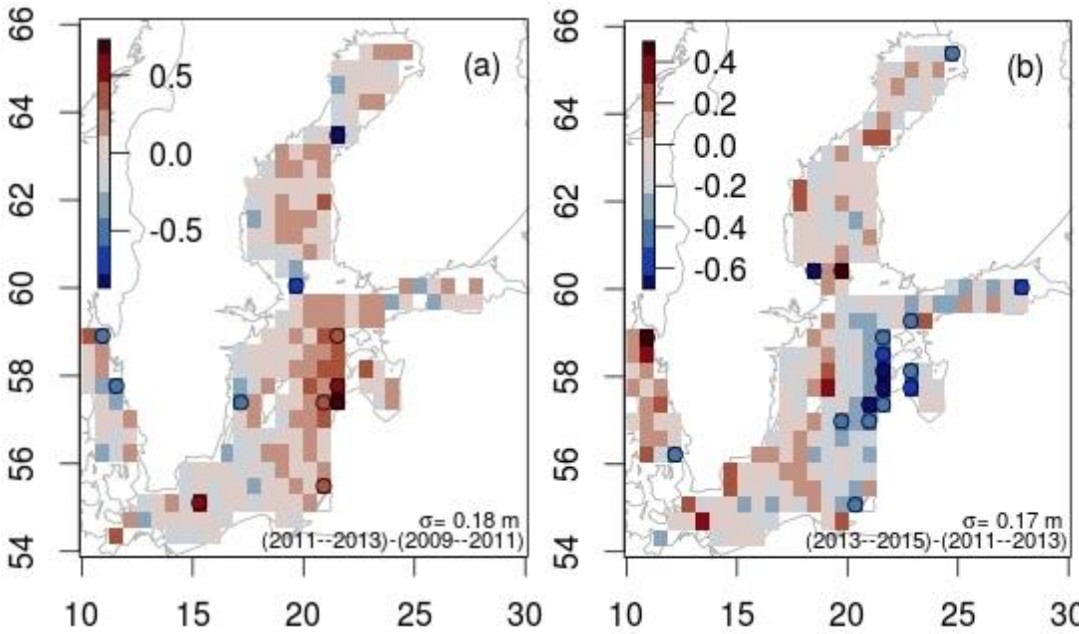

**Figure 6: Local spatial variability of wave climate in the Baltic Sea, (a) Changes (m) in the average SWH from 2009–2011 to 2011–2013, (b) Changes (m) in the mean SWH from 2011–2013 to 2014–2015. Crosses indicate the pixels hosting a statistically significant (at a >95 % level) increase or decrease.**

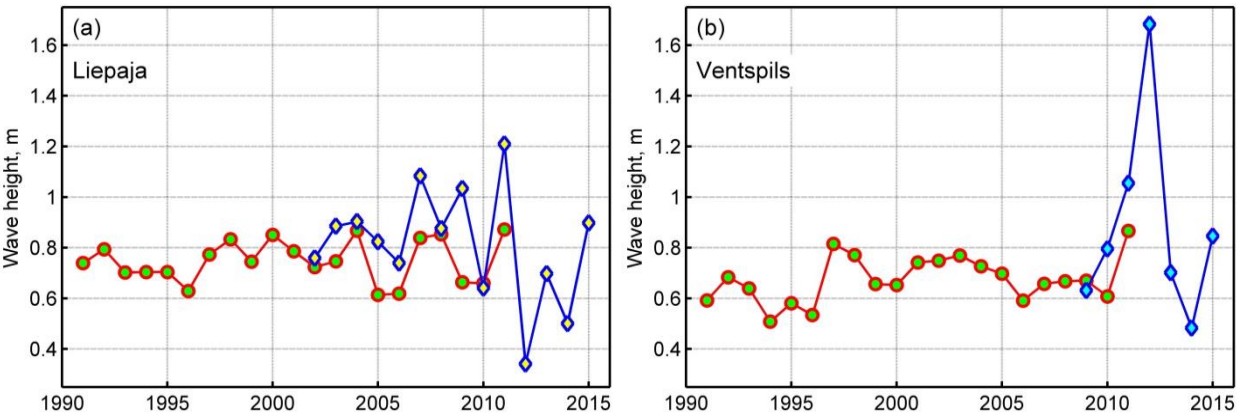

**Figure 7: Comparison of average annual SWH obtained using visual observations (circles) at Liepaja (a) and Ventspils (b) and multi-mission altimetry data (diamonds, shifted down by 0.4 m to ease a comparison).**

Table 1. Description of satellite missions used in the paper. As the number of observations made by the Topex/Poseidon mission was small it was not possible to adequately validate these data (Kudryavtseva & Soomere 2016), the relevant data set is not used in the analysis presented in this paper.

| Satellite | Period of observations | Accuracy [m] | Repeat cycle [d] |
|---|---|---|---|
| GEOSAT | 1985–1989 | 0.10 | 17.05 |
| | 2000–2008 | 0.10 | 17.05 |
| ERS-1 | 1991–1996 | 0.05 | 3/35/168 |
| TOPEX | 1992–2005 | 0.02 | 9.9156 |
| POSEIDON | 1992–2002 | 0.02 | 9.9156 |
| ERS-2 | 1995–2004 | 0.03 | 35 |
| ENVISAT | 2002–2012 | 0.03 | 35 |
| JASON-1 | 2002–2013 | 0.02 | 9.9156 |
| JASON-2 | 2008–2015 | 0.02 | 9.9156 |
| CRYOSAT-2 | 2010–2015 | 0.05 | 369 |
| SARAL/ALTIKA | 2013–2015 | 0.02 | 35 |