# Peer review of "Satellite altimetry reveals spatial patterns of variations in the Baltic Sea wave climate"

_Earth System Dynamics, 2016_

## Referee Comment (RC1) · Anonymous Referee #1 · 10 Jan 2017

Article "Satellite altimetry reveals spatial patterns of variations in the Baltic Sea wave climate " by Nadezhda Kudryavtseva and Tarmo Soomere adress relevant scientific questions within the scope of EDS. It's present novel concepts in to the long term wave climate research at the BS.

Are substantial conclusions reached? YES Are the scientific methods and assumptions valid and clearly outlined? Yes Are the results sufficient to support the interpretations and conclusions? Is the description of experiments and calculations sufficiently complete and precise to allow their reproduction by fellow scientists (traceability of results)? Do the authors give proper credit to related work and clearly indicate their own new/original contribution? YES Does the title clearly reflect the contents of the paper?

YES Does the abstract provide a concise and complete summary? YES Is the overall presentation well structured and clear? I was surprised, but YES Is the language fluent and precise? YES

Are mathematical formulae, symbols, abbreviations, and units correctly defined and used? I MORE PREFER TO USE WAVE HEIGHT IN CM, HERE IT IS PRESENTED IN M. BUT THAT IS OPTIONAL.

Should any parts of the paper (text, formulae, figures, tables) be clarified, reduced, combined, or eliminated? NO

Are the number and quality of references appropriate?

YES Is the amount and quality of supplementary material appropriate? YES

---

## Referee Comment (RC2) · Anonymous Referee #2 · 8 Feb 2017

**Review of  Kudryavtseva and Soomere: Satellite altimetry reveals spatial patterns of variation in the Baltic Sea wave climate**

Manuscript "Satellite altimetry reveals spatial patterns of variation in the Baltic Sea wave climate" by Kudryavtseva and Soomere utilises satellite altimeter data to estimate spatial and temporal variations in the significant wave height in the Baltic Sea. This is interesting work, since it is the first time altimeter data has been this extensively used to evaluate the wave conditions in the Baltic Sea. The Authors have evaluated the representativeness of their dataset and studied the statistical significance of the results. Earlier studies are cited and discussed in many places, but the comparison to them and the related discussion should be more comprehensive, especially if the reliability of the earlier studies is questioned. And as the altimeter dataset has been thoroughly validated, it should be indicated, whether the significant findings (trends, spatial patterns) are in the area, where the accuracy of the altimeter data was found to be adequate and if not, how it affecst the results and their interpretation. I also suggest that some of the figures would be redrawn to better illustrate the content and support the conclusions made by the Authors. Furthermore, colour scales that better describe the differences and are able to show where the differences are significant between adjacent pixels (for example from cmocean)  would better emphasize their content and make their interpretation easier.

**Some specific comments:**

**Abstract**

The abstract could better reflect the content of the paper.  The validation of the dataset is not main point in this paper, since it is already published. And is it necessary to name some of the satellites in the abstract? Maybe some more information should be presented about the reliability of the trend and the spatial patterns, since they are major part of this study.

**Introduction**

Has satellite altimeter data been used to evaluate the wave conditions in other regional seas? The references in the introduction (Hermer et al., Young et al.) are focusing more on global oceans or large domains.

**Data and methods**

Could the information about the satellites used in this study be presented in a table (name of satellite, period of measurements, coverage, accuracy, …). It is difficult to gather this information from the text, since its scattered to many places.

Page 2, line 21. Here it is said, that there is data from nine satellites, but in the abstract the Authors state that data from ten satellites was used.

Page 3, lines 5 -8. Expressions 'less convincing' and 'poorer' are quite general in nature and do not tell the reader, whether there is bias, or larger scatter in the comparison against in-situ data from smaller sub-basins or nearshore location. Although a comprehensive validation is presented in the earlier paper, the summary here should have enough facts (as numbers or with more precise expressions) to give a more comprehensive picture of the altimeter data quality in different areas of the Baltic Sea.

Page 3, lines 27-28. Which ice product is used to estimate the ice concentration in the Baltic Sea and what is the accuracy of this product? And how it affects the interpretation of results in areas,

where there is large variation in the extent of seasonal ice cover?

Page 6, line 23. 0.3 degree grid cell ( c. 33 km) is quite large considering comparison against coastal visual observations. Have you considered using smaller grid cells, additionally to the present comparison and analysis, or simply selecting the nearest locations from the altimeter data to the locations of the visual observation.

Page 6, lines 25- 28. The number of match-up pairs in Liepaja and Ventspils should be added. Although, there is a big difference in the number of observations between the early and later years of satellite mission, the tracks and coverage of the satellites are different and it is not clear how this reflects to coastal areas, in which these comparisons are made.

**Results**

Page 4, lines 15-16. Is the mismatch of inter-annual variation between the present dataset and the datasets presented by Soomere and Räämet and Soomere et al.? It is not quite clear, whether the Authors refer here to the previous sentence or to something else. It is not also clear, why this signals the importance of ice information in wave modelling. Please be more precise in expression and provide analysis/comparison that support these claims.

Page 4, lines 17-18. "heavily questions the reliability of existing wave reconstruction" -  This is quite strong statement and what is its justification. More analysis on the reliability of this reconstruction should be presented before declaring all the existing ones as questionable.

Fig 4. The right panel representing the number of observations has some white cells in areas for which there is value presented on the left panel (for example in the Gulf of Finland and Gulf of Riga). If there are no observations for these areas (or does the white color indicate something else), how can the linear trends be calculated and presented?

Figures 2 and 4. Why is different pixel size used to present the mean values of significant wave height and the linear trends?

Page 5, lines 27-28. It is not clear to me from Fig. 4a that there is a strong spatial pattern. And which smaller sub-basins are meant here? It seems . Maybe the color scale is not optimal to support these findings (at least on my screen or in the printed paper version). To my eyes it looks that there are large variations in the whole sea area and also within the basins.

Fig 5b. It is difficult to interpreted the colours under the crosses. Is it necessary to have them in the Figure?

**Discussion and conclusions**

I would expect more discussion on whether the significant changes were found in areas, which showed good accuracy in the validation against in-situ data. How the 'less-convincing' match in the smaller sub-basins affects the interpretation of the results. And what about the coastal areas? They were reported to have 'poorer' quality in the validation. Comparison against visual observation was said to show fairly good qualitative correspondence, but the correlations were quite low.

Lines 18-20: To my understanding the Baltic Sea is rather small and therefore, there the effect of wind direction might be more important due to changes in fetch. Is this true for all regional seas?

---

## Author Comment (AC2) · 7 Mar 2017

We thank you for your review.
* * *

---

## Author Response (AR1)

Please find the reply of the authors below. The comments, questions, and suggestions of the referees are presented in italics. The response of the authors is in regular font, and implemented changes to the manuscript together with line numbers are marked in dark blue font.

**General comments**

*Earlier studies are cited and discussed in many places, but the comparison to them and the related discussion should be more comprehensive, especially if the reliability of the earlier studies is questioned.*

As differences between numerically simulated properties of wave climate have been thoroughly discussed in several recent studies [e.g. Nikolkina et al. 2014. Multidecadal ensemble hindcast of wave fields in the Baltic Sea, *The 6th IEEE/OES Baltic Symposium Measuring and Modeling of Multi-Scale Interactions in the Marine Environment, May 26–29, Tallinn Estonia*; IEEE Conference Publications, 9 pp., doi: 10.1109/BALTIC.2014.6887854; Hünicke et al. 2015. Recent change – sea level and wind waves. In: The BACC II Author Team, *Second Assessment of Climate Change for the Baltic Sea Basin.* Springer, 155–185; Soomere T. 2016. Extremes and decadal variations in the Baltic Sea wave conditions. In: Extreme Ocean Waves, Pelinovsky E., Kharif C. (eds.). Springer, 107–140], our intention was to focus on novel aspects of our research. However, we agree with the Referee that a wider reflection of matches and mismatches between the results of earlier studies will strengthen the paper, and we shall be happy to expand the relevant sections of the manuscript during revision.

An additional discussion was added to the manuscript (p.8, lines 27-33, p.9, lines 1-5).

*And as the altimeter dataset has been thoroughly validated, it should be indicated, whether the significant findings (trends, spatial patterns) are in the area, where the accuracy of the altimeter data was found to be adequate and if not, how it affects the results and their interpretation.*

Thank you for indicating this aspect. It is true that we implicitly assumed that the accuracy of altimeter dataset is basically the same in the entire interior of the Baltic Sea. This assumption is reasonable because most of in situ data used for validation are located relatively close to the shore, where the altimeter data can be easily distorted, and even then the in-situ data showed a very good correspondence with the altimeter measurements. The most reliable altimeter data are in the south-eastern part of the Baltic Proper. This region has a relatively straight coastline, almost no islands, the very infrequent presence of ice and large spatial extension. We intend to insert the proper explanation into the revised version of the manuscript.

An additional discussion about the satellite altimetry data quality was added to the discussion (p.8, lines 18-26) and data and methods section (p.3, lines 8-14).

*I also suggest that some of the figures would be redrawn to better illustrate the content and support the conclusions made by the Authors. Furthermore, colour scales that better describe the differences and are able to show where the differences are significant between adjacent pixels (for example from cmocean) would better emphasize their content and make their interpretation easier. Some*

Thank you; we shall carefully manage all figures for the revised manuscript.

The color scales of the plots were adjusted. The palette of Fig.2, Fig.4, Fig. 5, Fig. 6 was changed. Fig. 7 was redrawn to match the style of Fig. 3.

**Specific comments**

*Abstract. The abstract could better reflect the content of the paper. The validation of the dataset is not main point in this paper, since it is already published. And is it necessary to name some of the satellites in the abstract? Maybe some more information should be presented about the reliability of the trend and the spatial patterns, since they are major part of this study.*

The abstract will be adjusted as recommended. In particular, it will be made clear that we rely on already cleaned and validated dataset. We are not sure whether removing of all names of satellites is good, but we shall definitely work towards keeping only the core information and towards adding additional information about the reliability of the trends and spatial patterns.

The abstract was modified respectively, the names of satellites removed, added a sentence about a comparison with the visual observations (p.1, lines 9 - 16).

***Introduction***. *Has satellite altimeter data been used to evaluate the wave conditions in other regional seas? The references in the introduction (Hemer et al., Young et al.) are focusing more on global oceans or large domains.*

The use of satellite altimeter data for evaluation of wave properties in smaller semi-enclosed water bodies has been scarce. Some papers focusing on the usage of this information for the regional wave climate are mentioned on page 2, lines 16–18. During the review process, several papers addressing this topic have become available, and we, of course, will add the relevant references and discuss the presented results in the introduction of the revised manuscript. The set of such paper involves, e.g. (Patra & Bhaskaran 2016, Trends in wind-wave climate over the head Bay of Bengal region; Kong et al. 2016, Validation and application of multi-source altimeter wave data in China's offshore areas). Some work in this direction has been performed for the Arctic Ocean, namely (Stopa et al. 2016. Wave climate in the Arctic 1992–2014: seasonality and trends: the paper about changes in the Arctic wave climate based on satellite altimetry and modeling, and Liu et al. 2016 Wind and wave climate in the Arctic Ocean as observed by altimeters).

Additional references which recently appeared, are added to the manuscript (p. 2, lines 14-15, lines 19-20).

***Data and methods***. *Could the information about the satellites used in this study be presented in a table (name of satellite, period of measurements, coverage, accuracy, …). It is difficult to gather this information from the text, since it's scattered to many places.*

Thank you, this is a good idea. A table listing all satellites, names, the period of measurements, coverage, and accuracy will be added to the manuscript.

Table 1 is added to the manuscript (page 18), a reference to the table is added (p.2, line 24).

*Page 2, line 21. Here it is said, that there is data from nine satellites, but in the abstract the Authors state that data from ten satellites was used.*

The discrepancy appeared from the fact that there were only a few observations from the Topex/Poseidon mission. As it was not possible to adequately validate these data, the relevant data set was excluded from the analysis. This is discussed in the paper describing the validation of the dataset (Kudryavtseva & Soomere 2016). The text will be amended accordingly to make this aspect clear.

To lessen the confusion, we changed a number of satellites to 9 in the abstract (p.1, line 9).

*Page 3, lines 5–8. Expressions 'less convincing' and 'poorer' are quite general in nature and do not tell the reader, whether there is bias, or larger scatter in the comparison against in-situ data from smaller sub-basins or nearshore location. Although a comprehensive validation is presented in the earlier paper, the summary here should have enough facts (as numbers or with more precise expressions) to give a more comprehensive picture of the altimeter data quality in different areas of the Baltic Sea.*

This part will be re-written using more explicit expressions and more exact quantification.

The part was re-written, the difference between the data quality in the Baltic Proper and coastal areas is discussed in more details, added a sentence that the data very close to the shore showing not good data quality were excluded from the presented analysis (p.3, lines 8-14).

*Page 3, lines 27–28. Which ice product is used to estimate the ice concentration in the Baltic Sea*

*and what is the accuracy of this product? And how it affects the interpretation of results in areas, where there is large variation in the extent of seasonal ice cover?*

The ice concentration measurements (OSI-409-a dataset) were taken from EUMETSAT OSI SAF Global Sea Ice Concentration Reprocessing data [EUMETSAT Ocean and Sea Ice Satellite Application Facility. Global sea ice concentration reprocessing dataset 1978–2015 (v1.2, 2015). Norwegian and Danish Meteorological Institutes, http://osisaf.met.no]. A description of the product, its accuracy, and how the results can be affected by large changes in the variations of the ice cover will be added to the paper.

However, large variations in the extent of ice cover are an intrinsic and deeply nontrivial question in studies of wave climate of partially ice-covered sea areas. The presence of extensive ice cover may render the basic properties such as average wave height almost useless (Tuomi et al. 2011. Wave hindcast statistics in the seasonally ice-covered Baltic Sea, Boreal Environ. Res., 16(6), 451–472). We shall amend the text to cover this aspect as well.

A reference to the ice concentration dataset is added (p.3, lines 32-34, p.4 lines 1-5).

*Page 6, line 23. 0.3 degree grid cell (c. 33 km) is quite large considering comparison against coastal visual observations. Have you considered using smaller grid cells, additionally to the present comparison and analysis, or simply selecting the nearest locations from the altimeter data to the locations of the visual observation.*

We are particularly thankful for this question that has led us to the identification of a mistake in our script. Namely, in comparisons of the results of visual observations with those derived from satellite altimetry, the correlation coefficient mentioned in the discussion paper was calculated for a much larger area. Calculating the correlation coefficient for the 0.3-degree grid cell results in a correlation coefficient of 0.61 for Liepaja (9 years of coinciding data) and 0.81 for Ventspils (3 years of coinciding data). A reduction of the grid cell to 0.2 degrees almost does not affect the correlation coefficient for Liepaja (0.60, 9 years of coinciding data) and improves the formal correlation for Ventspils (0.93, 3 years of coinciding data). The further decrease in the grid cell size leaves too few data points. The parts of the manuscript presenting the comparison of the outcome of satellite altimetry with the visually observed wave properties will be changed accordingly.

The part describing a comparison with the visual observations is re-written, describing correlation coefficient and how the change in the grid cell size affects the correlation (p.7, lines 19 - 28).

*Page 6, lines 25–28. The number of match-up pairs in Liepaja and Ventspils should be added. Although, there is a big difference in the number of observations between the early and later years of satellite mission, the tracks and coverage of the satellites are different and it is not clear how this reflects to coastal areas, in which these comparisons are made.*

The number of match-up pairs in Liepaja and Ventspils will be added to the text of the revised version.

The number of pairs was added to the manuscript (p. 7, lines 19 - 21).

*Page 4, lines 15–16. Is the mismatch of inter-annual variation between the present dataset and the datasets presented by Soomere and Räämet and Soomere et al.? It is not quite clear, whether the Authors refer here to the previous sentence or to something else. It is not also clear, why this signals the importance of ice information in wave modelling. Please be more precise in expression and provide analysis/comparison that support these claims.*

The comparison of the dataset presented in the manuscript and the wave modeling results discussed in Soomere and Räämet will be discussed in more detail, and the importance of ice in such comparisons will be clarified in the revised version.

The part was re-written to discuss in more details results presented in Soomere and Räämet (2014) (p. 4, lines 30-32, p.5, lines 1-5).

*Page 4, lines 17–18. "heavily questions the reliability of existing wave reconstruction" – This is*

*quite strong statement and what is its justification. More analysis on the reliability of this reconstruction should be presented before declaring all the existing ones as questionable."*
We will rephrase this phrase.
The sentence is modified accordingly (p.5, lines 5-7).

*Fig 4. The right panel representing the number of observations has some white cells in areas for which there is value presented on the left panel (for example in the Gulf of Finland and Gulf of Riga). If there are no observations for these areas (or does the white color indicate something else), how can the linear trends be calculated and presented?"*
Some points appeared white due to a problem with a color scale in the plotting software. This bug is fixed now. The plot with the number of observations will be changed to the correct one.
The color scale was changed (p. 16, fig. 4b).

*Figures 2 and 4. Why is different pixel size used to present the mean values of significant wave height and the linear trends?*
The linear trends are calculated for the annual mean values, whereas the average significant wave height figure is plotted using all available data over the whole period of observations. The decrease in the pixel size was a compromise to get a reasonable number of data points for each year for each particular pixel.
The pixel size for linear trends and mean significant wave heights were not modified.

*Fig 5b. It is difficult to interpret the colours under the crosses. Is it necessary to have them in the Figure?*
The crosses show statistically significant trends. Therefore, removing them might mislead a reader. We will change the symbol to a dot, so it will be easier to recognize and interpret the colors.
The crosses in Fig. 4a, 5b, 6a, 6b were changed to circles.

***Discussion and conclusions**. I would expect more discussion on whether the significant changes were found in areas, which showed good accuracy in the validation against in-situ data. How the 'less-convincing' match in the smaller sub-basins affects the interpretation of the results.*
*We will add a more detailed discussion about the changes in the wave heights in the areas with very good accuracy in satellite data and discuss more the results in coastal zones and smaller sub-basins. And what about the coastal areas? They were reported to have 'poorer' quality in the validation. Comparison against visual observation was said to show fairly good qualitative correspondence, but the correlations were quite low.*
As indicated above, we had a bug in the script for evaluation the correlation coefficients between the outcome of satellite altimetry and visual observations. The relevant parts of the text will be adjusted accordingly; in particular, we can speak about very good correspondence, with correlation coefficients >0.6.
An additional discussion about the satellite altimetry data quality was added to the discussion (p.8, lines 18-26).

*Lines 18-20: To my understanding the Baltic Sea is rather small and therefore, there the effect of wind direction might be more important due to changes in fetch. Is this true for all regional seas?*
Thank you, this is a very good observation, and we have tried to stress this aspect in several earlier publications. The strong dependence of wave properties on wind direction in the Baltic Sea mostly stems from the elongated shape of the Baltic Sea and less from its size. However, it is, of course, true that changes in strong wind directions will lead to large variations in the location of highest waves in more regularly shaped basins. To make this aspect clear, we shall add a more detailed comparison of the observed changes in the Baltic Sea with the published changes in other regional seas.
The discussion section was modified (p.8, lines 27-33, p.9, lines 1-5)

[revised manuscript text omitted]